# Scour at Bridge Foundations in Supercritical Flows: An Analysis of Knowledge Gaps

**Oscar Link** [1], **Emmanuel Mignot** [2,*], **Sebastien Roux** [3], **Benoit Camenen** [4], **Cristián Escauriaza** [5], **Julien Chauchat** [6], **Wernher Brevis** [7] **and Salvatore Manfreda** [8]

1   Department of Civil Engineering, University of Concepción, Edmundo Larenas 215, Concepción 4030000, Chile
2   University of Lyon, INSA de Lyon, LMFA, 20 avenue Einstein, 69621 Villeurbanne CEDEX, France
3   Centre d'Analyse Comportementale des Ouvrages Hydrauliques (CACOH), Compagnie Nationale du Rhône, 69007 Lyon, France
4   Irstea, UR RiverLy, centre de Lyon-Villeurbanne, 5 Rue de la Doua, CS 20244, F-69625 Villeurbanne CEDEX, France
5   Departamento de Ingeniería Hidráulica y Ambiental, Pontificia Universidad Católica de Chile, Santiago 3580000, Chile
6   Institute of Engineering, University Grenoble Alpes, CNRS, Grenoble INP, LEGI, 38000 Grenoble, France
7   Departamento de Ingeniería Hidráulica y Ambiental and Departamento de Minería. Pontificia Universidad Católica de Chile, Santiago 3580000, Chile
8   Department of European and Mediterranean Cultures, University of Basilicata, via Lanera 20, 75100 Matera, Italy
*   Correspondence: Emmanuel.mignot@insa-lyon.fr

**Abstract:** The scour at bridge foundations caused by supercritical flows is reviewed and knowledge gaps are analyzed focusing on the flow and scour patterns, available measuring techniques for the laboratory and field, and physical and advanced numerical modeling techniques. Evidence suggests that the scour depth caused by supercritical flows is much smaller than expected, by an order of magnitude compared to that found in subcritical flows, although the reasons for this behavior remain still unclear. Important questions on the interaction of the horseshoe vortex with the detached hydraulic-jump and the wall-jet flow observed in supercritical flows arise, e.g., does the interaction between the flow structures enhance or debilitate the bed shear stresses caused by the horseshoe vortex? What is the effect of the Froude number of the incoming flow on the flow structures around the foundation and on the scour process? Recommendations are provided to develop and adapt research methods used in the subcritical flow regime for the study of more challenging supercritical flow cases.

**Keywords:** bridge foundations; scour; supercritical flows; sediment hydraulics

## 1. Introduction

A vast amount of research on scour at bridge piers and abutments (referred here to as bridge foundations) has been conducted in the past, mostly focusing on results obtained from flume experiments with subcritical flow conditions, i.e., with a Froude number smaller than 1 and sand as a bed material. Even though some issues are still unresolved, the current knowledge has enabled the development of a number of guidelines for bridge design in different countries, e.g., HEC-18 in the US [1], in New Zealand [2], DWA-M529 in Germany [3], and the Ministry of Public Works in Chile [4], among others. However, there is an important lack of knowledge in transferring these methodologies and theories to bridge foundation design when they are placed in rivers with supercritical flow conditions.

Large-scale supercritical free-surface flows can occur in different environments. Some examples can be found in cases of flooded urban streets, fish-ways, tsunami inland flows, coastal channels, and mountain rivers. This paper focuses on the flow and scouring patterns at bridge foundations in rivers with supercritical conditions. The occurrence of supercritical flows in rivers is defined by high longitudinal slopes (>1%) and/or rapid flood waves. Commonly, steep rivers present gravel beds or mixtures of fine and coarse sediments, containing all possible sizes, from clay and silt up to boulders tens of centimeters in size. In dentritic networks, streams with a low Strahler's order (i.e., <3) are steep and produce flash floods but normally possess a small cross-sectional width. Therefore, deck bridges without foundations in these riverbeds are usually selected. At piedmont, however, rivers widen and it is common to observe cross sections with widths over 50 m, where bridge foundations may have to be included. Salient examples of such configurations are often encountered in steep watersheds subjected to heavy rains, such as on the Panamericana Route along Perú and Chile, La Réunion Island (Indian Ocean) in Taiwan or Japan, and also in a few European Alpine piedmont rivers (Figure 1). The examples in Figure 1 clearly highlight that supercritical flows are associated with a significant amount of energy for scouring and dynamic loading of the superstructure. Wood debris can also enhance the risk of pier stability. Such flows thus produce among the worst hydraulic conditions for bridge design. A recent bridge collapse due to scour in a supercritical flow occurred at the Rivière Saint Etienne in the La Réunion island due to cyclone Gamède. This bridge, which connected a road with traffic of 65,000 vehicles per day, collapsed and modified the terrestrial transport route for a long time (Figure 1f,g), thereby producing large economic losses. This event motivated, in France, the funding of specific experimental studies on flow and scour patterns around bridge foundations in supercritical flows, thus opening a new line of research. This event also evidenced an important lack of knowledge, with implications for the hydraulic design of bridge foundations in many regions of the world where supercritical conditions occur. Indeed, high flow velocities, along with high sediment transport and turbidity, rapid changes in the local morphology, and air entrainment make it complex and sometimes impossible to perform flow and scour measurements in the field [5] or even in laboratory facilities [6].

In this paper, we identify the knowledge gaps in scour at bridge foundations in rivers with supercritical flows. These problem areas can be summarized as knowledge gaps in flow dynamics, past obstacles in flat and scoured beds, and scour patterns and mechanisms. This paper also reviews the applicability and limitations of the existing methodological approaches typically used in subcritical flows and sand beds, including field and laboratory measuring techniques for flow and scour, as well as physical and numerical modeling techniques. The link between experimental or numerical work at the local process scale and the long-term river dynamics that finally determine bridge failures is also highlighted. In most cases, we are forced to start with a well-studied case of scour in sand caused by a subcritical flow to provide a referential basis. The paper concludes with final remarks on the results of our analysis.

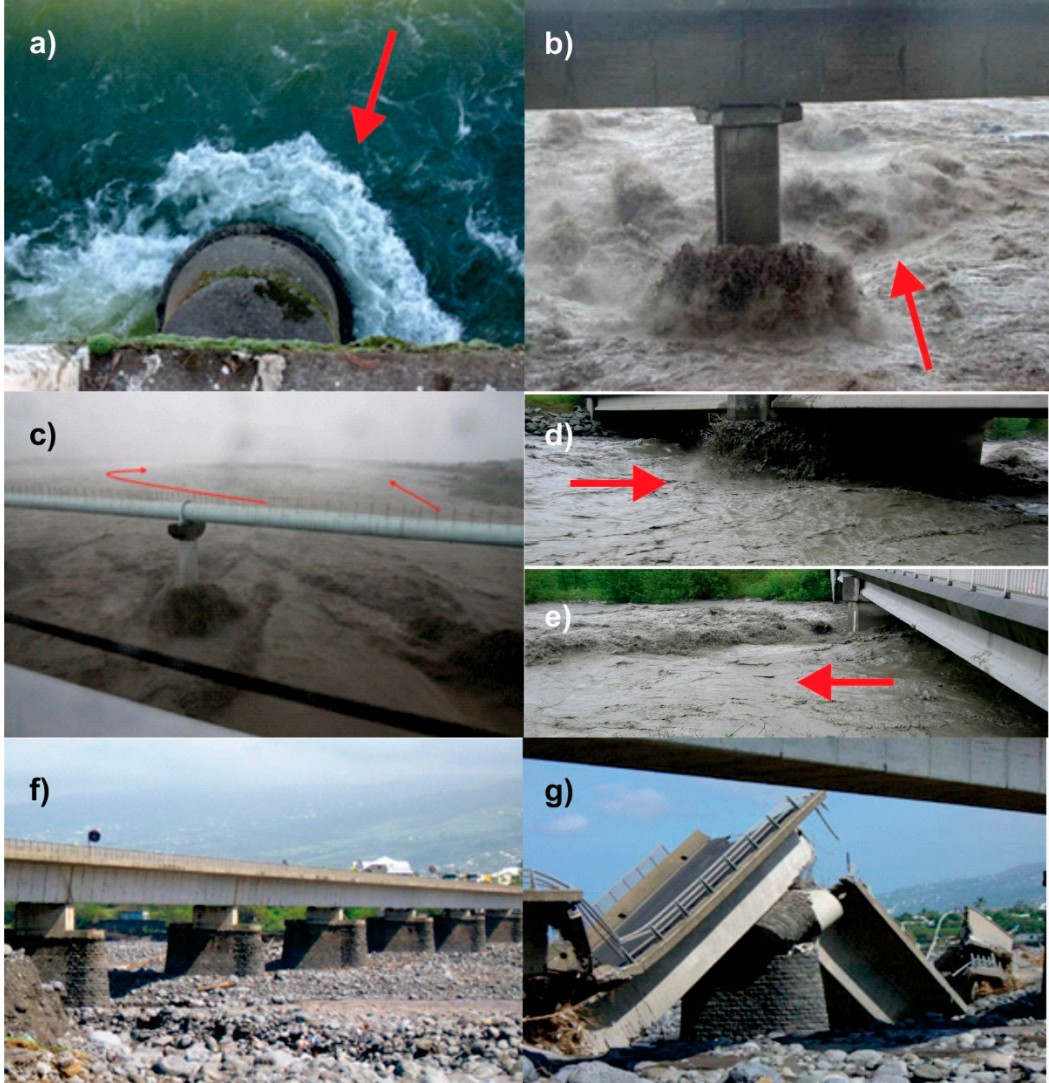

**Figure 1.** Photographs of supercritical flows at bridge piers in the Biobío River, Chile (**a**), La Rivière des Galets, La Réunion island (**b**), The Choshui River, Taiwan ((**c**) [7]), and the Arc-en-Maurienne River, France ((**d**) and (**e**) [8]). Photographs of the bridge at Rivière St-Etienne (La Réunion island) before (**f**) and after (**g**) the 2007 collapse ((**g**) [9]). Red arrows indicate the flow direction.

## 2. Flow Patterns around Bridge Foundations

Scour at bridge foundations involves complex interactions between the three-dimensional, unsteady, and turbulent flow, and the movable riverbed. The flow patterns around foundations have been described for cylinders mounted on plane beds, as well as for cylinders in scoured holes in both sub and supercritical flows. Important differences in the flow field around bridge foundations can be highlighted between these two flow regimes.

### 2.1. Surface Mounted Emerging Obstacles on Flat Beds

### 2.1.1. Subcritical Flows

Because of the adverse pressure gradient induced by an obstacle mounted on a flat surface, the incoming flow separates at the upstream junction of the obstacle and the bed, reorganizing into a complex large-scale dynamically-rich coherent structure, known as a horseshoe vortex (HSV) system, which wraps around the front and the flanks of the obstacle [10,11]. The turbulent HSV system presents a bi-modality of the probability density functions of velocity and pressure fluctuations with two

dominant modes called the backflow and the zero–flow modes [11,12]. The study in [13] documented the presence of a third mode, the intermediate mode, which is described as a mode close to the zero-flow mode, but with less intensity in the vertical velocity component of the near-wall jet [14–16]. The main HSV interacts with larger structures, impinging the separated region from upstream, as well as with hairpin vortices that develop underneath the horseshoe vortices [12,17,18]. The stagnation pressure causes an additional bow wave at the upstream free surface, adjacent to the obstacle, which, in the upstream symmetry plane, rotates in a direction opposite to the HSV. The stagnation pressure also causes a sideward acceleration of the flow at the sides of the cylinder [19].

### 2.1.2. Supercritical Flows

The study in [20] recently showed that the flow pattern around an obstacle in a supercritical flow varies as a function of the obstacle width ($D$) to flow depth ($h$) ratio, distinguishing the following patterns:

The detached hydraulic jump pattern: at high ratios of obstacle width to flow depth, i.e., $D/h$ > 0.5–2.0 a bow-wave like, detached, hydraulic jump takes place in front of the obstacle [21]. From up to downstream, the flow first slows down to pass from the supercritical to subcritical regime and eventually stops at the upstream face of the obstacle. This transition from the super- to sub-critical regime takes place through the detached hydraulic jump wrapping around the obstacle (Figure 2a). The foot of the hydraulic jump then follows a hyperbolic curve in the horizontal plane. Near the obstacle, the flow is in the subcritical regime, so a horseshoe vortex occurs in the near-bed region at the foot of the foundation. The foot of the horseshoe vortex then follows an elliptic curve [21]. Moreover, the separation distance between the obstacle and both the detached jump and the horseshoe vortex appears to increase with the non-dimensional flow depth ($h/D$) and decreases with the Froude number of the incoming flow [22].

The wall jet pattern: at small ratios of foundation width to flow depth (i.e., $D/h$ < 0.5–2.0) a so-called wall-jet-like bow wave develops (Figure 2b; [20]). The flow remains in the supercritical regime (unaffected by the presence of the obstacle) until reaching the foot of the obstacle. There, it deviates and goes up along the upstream face of the obstacle and slightly towards its sides, where it is evacuated and falls down in the flow further downstream. Part of the up-going flow rolls backward and falls down at the foot of the obstacle in periodic reverse spillage.

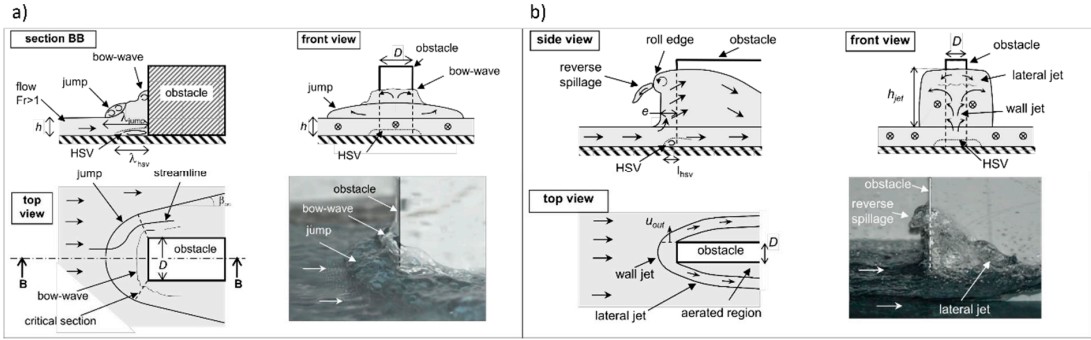

**Figure 2.** Flow patterns upstream of a rectangular emerging foundation in supercritical flow: (**a**) detached hydraulic jump and (**b**) wall jet pattern (adapted from [20]). HSV, horseshoe vortex.

### 2.2. Foundations with Scour Hole

### 2.2.1. Subcritical Flows

The HSV system around a scoured foundation presents similar patterns to systems around a cylinder on a plane bed, namely bimodal near wall velocity distributions, sizess, and frequencies that scale with the Reynolds number [18,23,24]. Turbulence properties are also reported by [25,26].

#### 2.2.2. Supercritical Flows

To the best of our knowledge, no velocity measurements have been carried out in the scour hole of a foundation in a supercritical flow. The only available information are flow depths measured under laboratory conditions by [27]. These measurements were obtained using a clear-water turbulent inflow with mobile bed conditions and $D/h \approx 10$. In the presence of a scour hole, a detached hydraulic jump wrapping around the abutment was observed, as in the plane bed case, but with the detached hydraulic jump closer to the obstacle. Additionally, two parallel bow waves smaller elevations were also visible downstream from the obstacle (Figure 3).

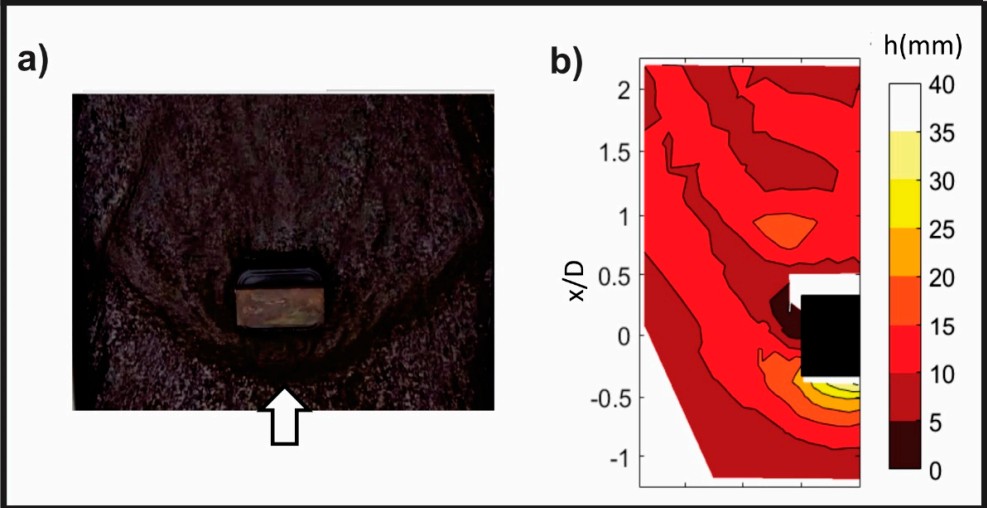

**Figure 3.** 2D field of water level around a scoured rectangular abutment in a clear-water supercritical flow by [27]. (**a**) Photograph at initial stage of scour, and (**b**) measured flow depth field at the quasi equilibrium stage of scour.

### 3. Scour around Bridge Foundations

Comprehensive studies on the variables controlling the maximum scour depth have been conducted since the state-of-the-art paper by [28]. The available information mostly corresponds to the case of scour in sand, with few exceptions for cases of scour in fine sediments [29–32] and gravel bed rivers [33–38]). Further, researchers have concentrated on the dynamics of scour [39–42]. A detailed description of the functional relationships between scour and its controlling variables [43,44] are out of the scope of the present paper. In this section, we focus on scour patterns.

#### 3.1. Subcritical Flows

The researchers in [45] conducted an experimental study of clear water scouring around a circular cylinder, identifying the HSV as the main scour mechanism, due to the enhanced bed shear stress under the horseshoe vortices. The authors in [46] used numerical simulations of the flow field in the scour hole around a cylinder to show the importance of side slides in the scouring process. The study in [47] showed differences in the scour mechanism, depending on the flow intensity ($I$), i.e., the ratio between the flow velocity and the critical velocity for the incipient motion of sediment particles. From the minimum scour formation threshold at $I \approx 0.4$ to 0.6, to the incipient motion condition at $I = 1$, clear-water conditions dominate, and the HSV and downflow produce sediment entrainment and landslides, which enlarge the scour hole over time. For intensities of $1.0 < I < \approx 4.0$, a bedload occurs, and fluctuations of the scour depth in time can be observed due to the bedforms entering the scour hole. For $I > 4.0$, entrainment into suspension occurs at the undisturbed bed, and refilling of the scour hole due to deposition is expected during the falling stage of floods. The study in [48] described the morphological evolution of dune-like bed forms downstream of bridge piers and abutments

that are generated by local scour. Recently, the authors in [49] conducted experiments that clearly distinguished two different scour modes: one, at the onset of erosion arising at the base of the cylinder and usually ascribed to the wrapping horseshoe vortex, which was determined and rationalized by a flow contraction effect, and another one, visible downstream of the cylinder, which consists of two side-by-side elongated holes. This pattern is observed for flow regimes close to the horseshoe scour onset, whose growth usually inhibits its spatio-temporal development.

### 3.2. Supercritical Flows

The study in [50] measured the maximum scour depths at piers in live-bed experiments for several flow configurations, including seven flows in the supercritical regime (Figure 4). For these flow conditions, no major increase in the maximum scour depth was reported as the Froude number exceeds 1.

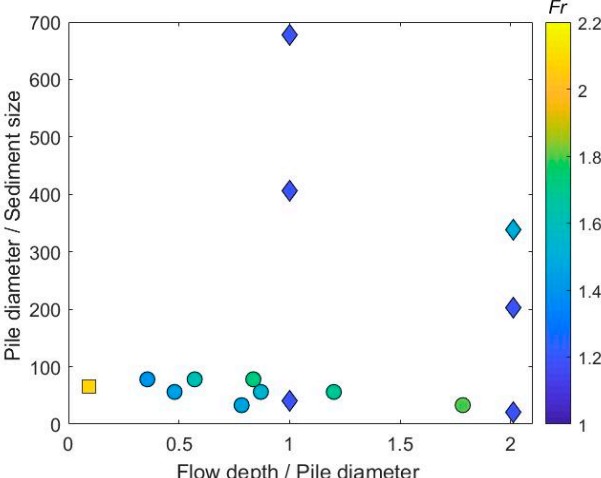

**Figure 4.** Measured maximum scour depth for flows in the supercritical regime: available configurations in the literature, with *Fr* as the Froude number.

The authors in [27] reported on scour at an abutment ($h/D = 0.095$) in a supercritical flow with clear-water conditions. The flow pattern was of the detached hydraulic jump type. The scour hole extended far downstream from the abutment sides (Figure 5a). Two symmetric deposition zones were observed downstream, separated by a streamwise valley of almost zero deposition elevation, with a magnitude of maximum deposition elevation about half the maximum erosion measured in the scour hole. The maximum scour depth at equilibrium was observed along the nose and the upstream part of the lateral faces of the abutment (see Figure 5b). Interestingly, the temporal evolution of the maximum scour depth agreed well with the scour formula in [51] for subcritical flows, when considering the conjugate depth and velocity downstream the straight hydraulic jump (see Figure 5c) as input flow parameters. More experimental evidence for the extrapolation of the results in [27] is needed, as the researchers in [27] investigated only a single flow configuration, with a low Reynolds number (Figure 4).

The study in [6] investigated scour at a pier or abutment in a supercritical flow using a 1:50 physical model of the projected bridge on the Rivière des Galets, located in the CNR (Compagnie Nationale du Rhône) laboratory (Figure 5d) to assess the shape and depth of the scour hole for different foundation diameters and approaching flow conditions. The ratio between the flow depth and pile diameter ranged between 0.4 and 1.8 (Figure 4), resulting in a wall-jet or detached jump flow patterns. Incoming velocities ranged from 1 to 2 m/s under a steady flow regime. The model sediment was scaled geometrically and consisted of a non-uniform mixture of sand and gravel, with a mean diameter of 1.8 mm. Around 20 tons of sand were supplied during running experiments to keep the sediment bed in equilibrium. Unfortunately, experimental conditions (turbidity of water, bed load displacement,

high flow velocity) did not allow the researchers to measure the hydrodynamics in the scour hole. The measurements of the scour depth during the test were carried out using three rows of metal rods in a comb arrangement (see Figure 5f), while the equilibrium scour (see Figure 5g) was measured using an automatic tacheometer after drying the bed (Figure 5h). Figure 5i shows the maximum scour depths for a range of pier diameters and approaching flow velocities, with a flow intensity $I$ up to 4. The maximum scour depth after the experiments appears to vary between 0.9 and 2.5 times the pier diameter, demonstrating that the scour is not as high as expected from the extrapolation of the results from subcritical flows, even though the flow velocities are much higher.

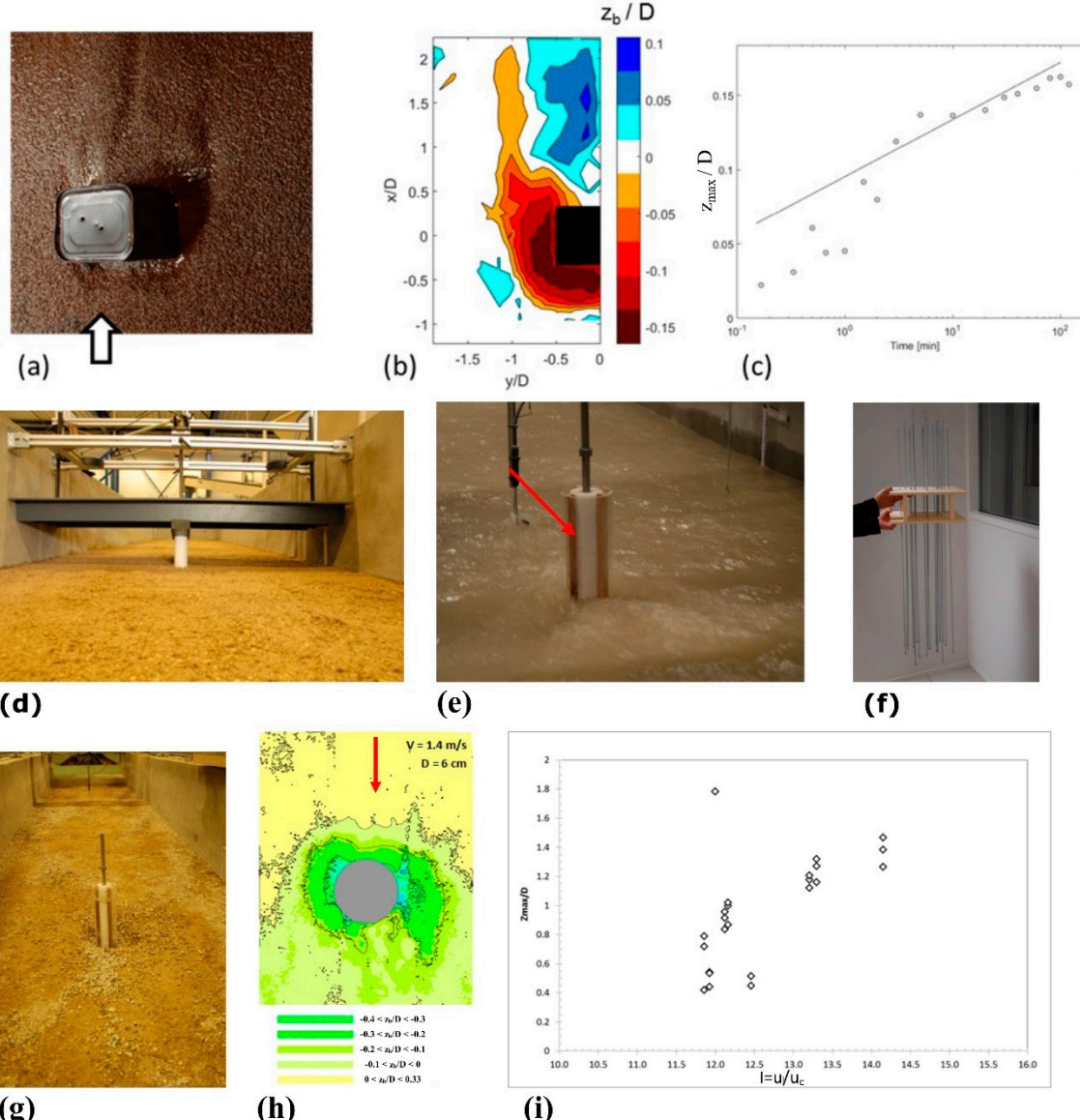

**Figure 5.** Clear-water scour at bridge foundations in experiments with a supercritical flow. Photograph of the clear water scour from [27] (**a**), with the corresponding normalized topography ($z_b$ being the bed elevation) at equilibrium (**b**) and the measured time evolution of normalized maximum scour depth $z_{max}$ (symbols) along with the prediction (plain line) obtained with the scour formula for subcritical flows [51] (**c**); live bed scour experiments from [6] (**d**), the flow pattern seen from downstream the pile (**e**), the metal rod comb used to measure the scour hole during the experiment (**f**), the scour shape at quasi-equilibrium after 2 hours at a pile of 6 cm in diameter in a 1.4 m/s flow condition (**g**, **h**), and the maximum normalized scour depth measured at the foot of the piles as a function of the flow intensity, $I$ (**i**).

## 4. Recommended Methods for the Study of Flow and Scour in Supercritical Flows

Future research should be devoted to better understanding the flow processes responsible for scour at bridge foundations in watercourses with a supercritical regime along with the quantification of the maximum scour depths. Recommendations on methods to develop research on scour in supercritical flows are given below for laboratory and field measurements, as well as physical and numerical modeling.

### *4.1. Laboratory Techniques for Flow and Scour Dynamics*

### 4.1.1. Standard Velocimetry

The use of micro high-speed propellers is appropriate when the quantification of local time average velocities is the measurement objective, such as in discharge measurements. However, the use of such propellers can be constrained by the presence of sediment due to the potential damage caused in the mechanical system. A modern approach for pointwise velocimetry is the use of Acoustic Doppler Velocimeters. Even though this instrument has been employed to measure the time averages and turbulent quantities in a broad range of flows, and even though the instruments have reduced dimensions, in cases of supercritical conditions, their intrusiveness may create a flow pattern similar to those caused by obstacles (Figure 2). Another consequence is the generation of oblique surface waves affecting the whole cross-section of the channel downstream.

### 4.1.2. Advanced Measurement of Hydrodynamic Processes

The use of advanced optical or acoustic measurement techniques for laboratory research in the supercritical regime presents many technical restrictions compared to subcritical cases. Supercritical flows, and particularly the flows developed around obstacles, show important unsteady free surface deformations, which make any attempt to access them from the surface impractical.

In the case of a flat smooth bed, measurements along the vertical planes can be performed through the bottom wall. One possibility is to use an acoustic profiler [52], located within the bed, which emits a vertical acoustic signal towards the free surface (Figure 6d). This instrument provides access to a time resolved vertical profile of three velocity components measured with a vertical resolution precision of approximately 1 mm. A second alternative is an optical technique using sourced illumination and non-orthogonal cameras located at the bottom of the channel. The most common optical technique is two-dimensional Particle Image Velocimetry (2D PIV) or 2D particle Tracking Velocimetry [53]. In this case, the reduction of optical distortions is achieved through the use of Scheimpflug adapters and optical prisms attached to the channel bottom-wall (as in standard stereoscopic or tomographic setups). A high image resolution and sampling frequency for the system becomes crucial for measuring the dynamics of small-scale structures around obstacles. A double pulse laser with a short-time interval between images is recommended to produce images with a high signal to noise ratio. An additional issue is air entrainment through the hydraulic jump or reverse spillage. In these cases, it would be necessary to consider the use of tracer particles, e.g., fluorescent particles, able to reflect the light source in a different wavelength than the light scattered by the air bubbles. This and additional band-pass filters in the camera give access to an independent quantification of particle and bubble dynamics. Finally, other alternatives available, using similar optical arrangements from the bottom, are the use of Stereo- and Tomographic-PIV ([54], Figure 5b), as well as the use of pointwise measurements performed with Laser Doppler Velocimetry [55].

The use of a movable bed constrains the use of optical measurements from the bottom. An alternative is to use refractive index-matching between the sediment and the fluid, as presented in [56], which would facilitate a similar technique to the one described above for the plane bed condition (Figure 6a).

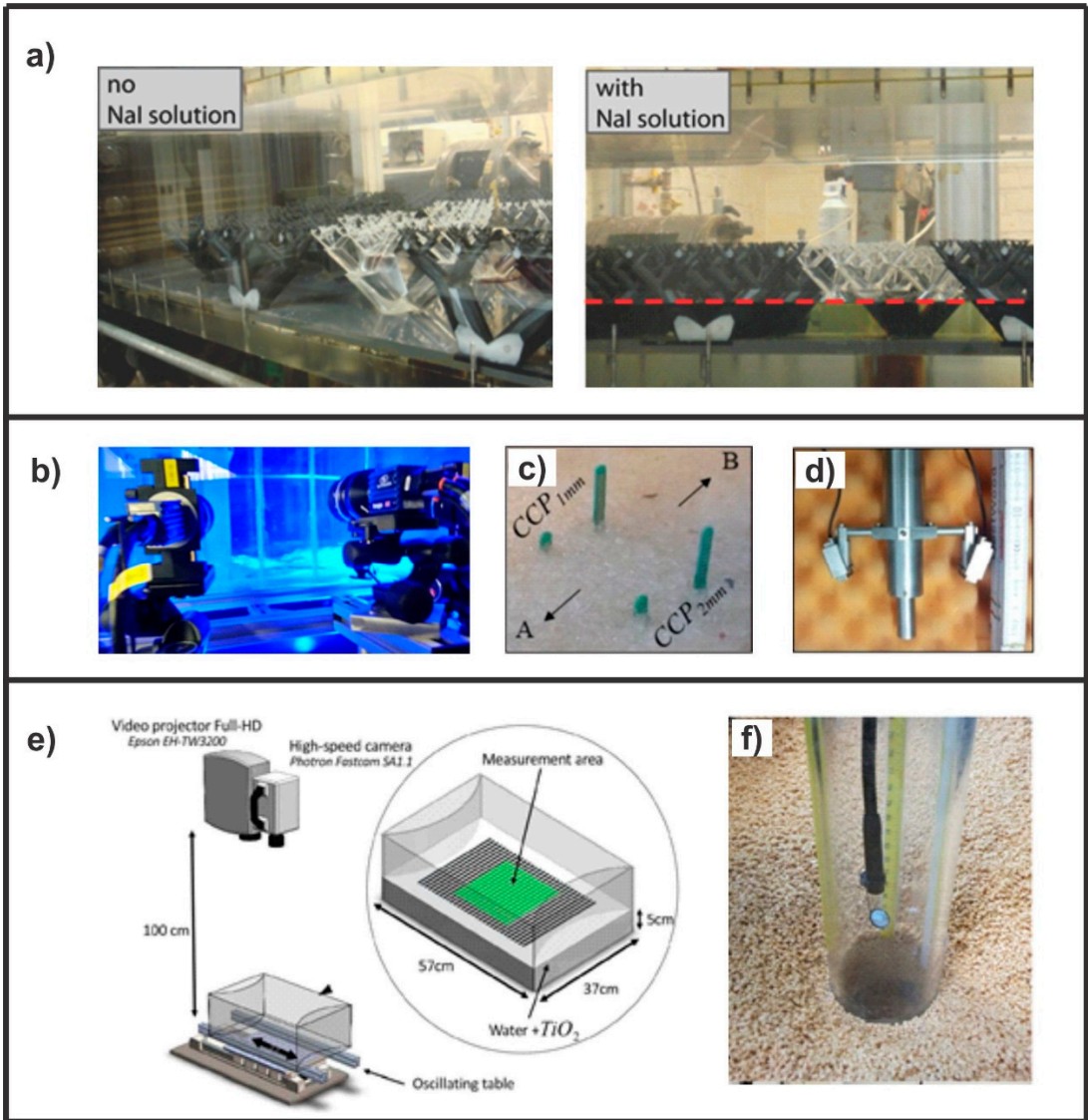

**Figure 6.** Innovative experimental techniques to be applied to supercritical hydrodynamics and scour processes in the laboratory: (**a**) an example of refractive-index and density matching from [57], (**b**) a view of a tomographic system, (**c**) CCP (Conductivity Concentration Profiler), (**d**) and ACVP (Acoustic Concentration and Velocity Profiler) from [58], (**e**) the application of Fourier transform profilometry to free-surface deformation, as per [59], (**f**) and an endoscopic camera placed inside a transparent pier for measuring scour evolution (from [47]).

### 4.1.3. Image-Based Reconstruction of a Free Surface

The time-resolved measurement of free-surface deformation is commonly performed using point-wise sensors, such as ultrasonic sensors (from above) or electrical resistive probes (within the water column). The study in [20] showed that these techniques apply to both supercritical flow patterns: the detached hydraulic jump and the wall-jet like bow wave. On the other hand, access to instantaneous 2D free-surface deformation around the pier/abutment requires projection techniques, such as the RGB-D sensor method [60], the projected grid method [27] (with a limited spatial resolution), or the Fourier transform profilometry method [61], with a higher spatial resolution (Figure 6e). These techniques require high quantum efficiency for the CCD and a high frequency sampling camera to get access to the dynamics of the surface deformation.

### 4.1.4. Distance Sensors for Scour Measurement

Laser distance sensors (LDS) are used to measure the scour-hole radius at different depths, thus quantifying the scour hole's geometry. In [62,63], researchers developed a measuring system composed of an LDS to measure the scour radius with an accuracy of ±0.4 mm. This LDS is placed inside a Plexiglas pier and aligned in a horizontal and radial direction, such that no refraction on the cylinder wall is observed during the measurements. The sensor is driven in the vertical direction by a step-motor with a precision of ±1/50 mm, and in the azimuthal direction, a vertical positioning system is driven by a second step-motor with an accuracy of ±1/100°, allowing the distance sensor to turn around in the scour-hole, taking various vertical profiles in different azimuthal half-planes. In this way, the geometry of the scour-hole below the original flat bed is automatically measured. The sensor performed well in tests with live bed conditions, having a flow intensity of $I = 2$. The same technique was applied to measurements in gravel [37] and sand-clay mixtures [30]. The application of LDS is, however, restricted to conditions close to clear-water and with low turbidity. It is, therefore, expected that LDS will not work properly in a number of supercritical cases where fine sediment is suspended.

Conductivity concentration profilers (see Figure 6c), originally developed in [64], were used to measure concentration profiles under sheet flow conditions in [58] using lightweight PMMA particles (1 and 3 mm). This technique is based on the inversion of the linear relationship between the sediment concentration and the conductivity of the medium. Using a grid of conductivity probes stuck on the pile, it would be possible to detect, at about 10 Hz, the position of the fixed bed interface at the pile position. Compared with LDS, the obtained information would be restricted to the bed elevation at the pile location, and no information on the 2D geometry of the scour hole would be provided. However, the advantage of such a technique is that it would be applicable even under intense live-bed conditions encountered in supercritical flow conditions for which no optical access is possible.

### 4.1.5. Image-Based Reconstruction of Scour during Running Experiments

The study in [65] presented a stereovision-based technique for continuous measurement of the bed morphology. This technique is capable of reconstructing instantaneous surface representations of the evolving bed with high spatial resolution during scour experiments. Two calibrated cameras must be partially submerged in the flow and record videos of the evolving bed geometry. This technique considers the texture of sediment beds and does not require the use of targets or structured light. A set of computer-vision and image-processing algorithms were developed for accurately reconstructing the surface of the bed. This technique was further applied to the spatio-temporal characterization of scour at the base of a cylinder in [66]. Optical access to the scoured region might be an issue in supercritical flows, due to noise from the unsteady wavy water surface. The authors in [67] developed a bed level tracking system with micro cameras placed inside a pier to record the maximum scour depth under clear-water conditions. The study in [47] used an endoscopic camera placed inside the pier to record images of a graduated pier, registering the maximum scour depth in scour experiments conducted under live-bed conditions (Figure 6f). Similarly, the authors in [68] used a snake camera of 0.5 cm in diameter that slightly penetrated the flow surface (less than 1 cm) to look at a graduated strip on the upstream face of the pier. In these studies, algorithms for the automatic recognition of the scour hole bottom through digital image processing were developed. The ability of cameras placed inside the pier to record the maximum scour depth has emerged as a promising alternative for measuring the temporal evolution of the maximum scour depth at piers in supercritical flows. Indeed, surface waves and oscillations, as well as air entrainment occurring in supercritical flows, constrain the applicability of stereoscopic systems using cameras placed above the water surface. Recognition of the scour-hole bottom, and thus the detection of the maximum scour depth in the images, might be possible even in presence of suspended sediment particles because the color of the turbid water is different to the color of the bottom embedded foundation.

4.2.1. Approaching Flow Measurements

The main restrictions for typical flow measurements at large velocities are related to instrument intrusion in the flow. For supercritical flows, instruments can only be deployed from a fixed point (a bridge or cable way), and intrusive measurements generally become impossible above a flow speed of 5 m/s (for a current meter on a torpedo). Acoustic Doppler Current Profilers (ADCP), which are currently used in the field to measure flow fields, are limited to a relative velocity of a few meters per second between the apparatus and the river, making their use generally impossible for supercritical flows. Indeed, these instruments are highly sensitive to air entrainment around the apparatus. From our own experience, the result quality using ADCP starts to decrease above a flow speed of 3.5 m/s. It should also be noted that such high flows are often associated with high suspended load concentrations and woody debris [69], which make such a measurement very dangerous.

The non-intrusive counterpart instruments used to measure large flow velocities during flood events are radar velocimeters ([70]; Figure 7a) and video analysis systems, such as Large Scale Particle Image Velocimetry (LSPIV; [71]). The main limitation of these systems is that they can only provide a surface velocity, so complex 3D flows cannot be described. Moreover, radar velocimeters are not able to provide a flow direction. LSPIV has been applied successfully to measure the velocity field in rivers with high discharges [72,73]. However, a difficulty that appears close to bridge foundations in supercritical flows is that the water surface may become 3D due to standing waves or antidunes. Since surface velocity estimations are made on a plane, this type of surface could lead to significant errors. On the other hand, stereovision-based techniques could be of interest to describe the water surface of such rapid flows.

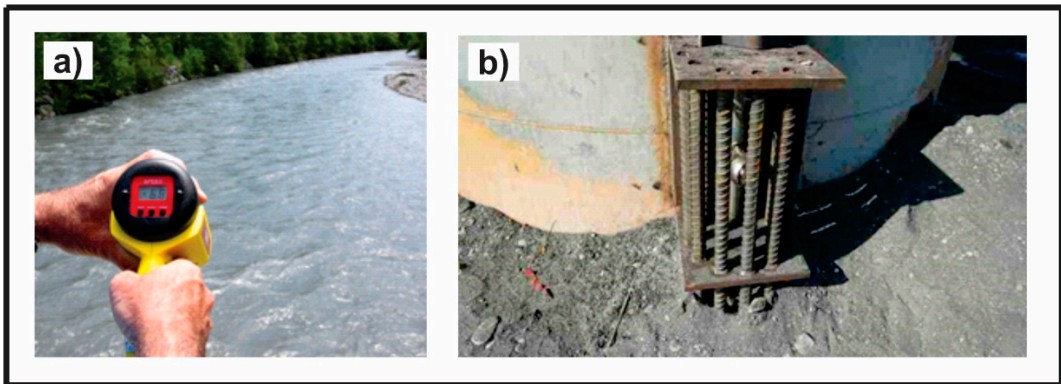

**Figure 7.** (**a**) Radar measurement from a bridge (Photo: P. Belleudy), and (**b**) a system for pier erosion monitoring used at NARLabs (National Applied Research Laboratories), Taiwan.

4.2.2. Bed Level Measurements

At present, only a few successful field measurements of scour during floods have been reported using the numbered-brick technique [5,74], which requires numbered bricks to be placed beneath the river bed by using excavators prior to the flood event. The location of the numbered-brick column can be accurately identified by using a total-station transit. After the flood has receded, excavators are again deployed to dredge the location of the preinstalled bricks to determine the number of bricks that have been washed away during the flood. To this end, one can then evaluate the maximum scour depth at the measuring location based on the remaining number of bricks [75]. Monitoring techniques for scour were reviewed in [76]. Such techniques include poles or sounding weights (with intrusion into the flow), float-out devices and tethered buried switches that can detect scour, and rods buried into the streambed (coupled with various sensors to measure the scour depth). These instrumentations may be robust, but they are limited to a single location and require expensive installation and/or maintenance.

Other techniques use sonars, fathometers (acoustic waves), or radars (electromagnetic waves) that are able to locally monitor the scour depth over time but are often limited to low flow velocities, water depths, and/or Suspended Sediment Matter (SSM) concentrations. Ground Penetrating Radars (GPR) allow a spatial view of the erosion but require manual operation and thus cannot be used during heavy-flood flows. Lastly, accelerometers provide a response for the bridge structure to scour. This structural dynamic response is able to provide a global view of the scour, which is of interest for early warning systems but is sensitive to the shape of scour holes and soil type and thus needs a site-specific calibration before monitoring [77].

As a consequence, a combination system based on structural dynamic response together with float-out or buried-typed devices would be the best actual monitoring approach to provide a quantitative description, over time, of the erosion depth at the foot of a bridge foundation in a supercritical flow.

Recent developments on active tracers [78] may allow their use around structures during a flood. Indeed, such systems could provide interesting results on gravel path and velocity but would be difficult to deploy during extreme events, such as those responsible for supercritical flows.

### 4.3. Physical Scale Models of Scour at Bridge Foundations

The authors in [79,80] analyzed the scale effects on scour, concluding that the use of laboratory flumes in developing accurate predictors of scour depth at full-scale piers is limited due to scale effects that may produce greater scour depths in the laboratory than at actual piers in rivers. The authors in [81] discussed the effects of sediment size scaling on the physical modeling of a bridge pier scour. Moreover, for model sediments smaller than d = 0.05 mm cohesion will reduce the scour [30]. Typical Froude scale models do not necessarily simulate the tractive forces and sediment erosion accurately because Froude scaling does not simulate viscous forces. Recently, researchers [82] showed that clear-water and live-bed scour under steady and unsteady flows in a subcritical regime is similar in the prototype and the model if the dimensionless flow work (as proposed in [83]) and the dimensionless grain diameter D* are equal in both (the prototype and the model). Still, verifications of these scaling laws for supercritical cases are needed before such a scaling approach can be employed with confidence in physical modeling.

Additionally, important difficulties in the physical modeling of live-bed scour caused by a supercritical flow are the high water and sediment discharges to be supplied as upstream boundary conditions. While the water discharge may be overcome with large pumps, the sediment supply is a challenge in itself. The researchers in [6] supplied up to 400 l/s of water in order to reach their 2 m/s target velocity and 3 kg/s (10.8 t/h) of sediment during their experiments, which lasted up to 2 hours (Figure 5d–i). Moreover, the sediment needed to be regularly removed from the downstream basin in order not to generate backwater curves from downstream and keep the supercritical flow condition in the flume. Flow velocities were measured with propellers and ADV (Figure 5e), ensuring that the system did not produce important disturbances of the flow conditions. Because very loaded flows are prevented from using sonar or optical measurement systems (Figure 6f), the most adapted scour measurement methods appeared to be classical thin rods systems deployed from above the free surface (Figure 5f).

### 4.4. Advanced Numerical Tools for Simulation of Scour in Supercritical Flows

Numerical simulations play a significant role in the understanding of the flow hydrodynamics and scour around hydraulic structures, as they can explore conditions that cannot be reproduced experimentally or are inaccessible for measurement devices, complementing the observations, and providing additional insights into sediment-flow interactions and feedback [84]. In the following, numerical simulation techniques to be applied to supercritical scour processes are discussed, including the coupled hydrodynamics and sediment transport approach and the multi-phase flow approach.

### 4.4.1. Coupled Hydrodynamics and Sediment Transport Approach

First numerical simulations of scour around a bridge pier under a subcritical flow regime were performed about 25 years ago in [85], using the Reynolds-Averaged Navier–Stokes (RANS) equations for the hydrodynamics, coupled with the Exner equation for the bed's morphological evolution in steady flow conditions. Another important milestone was achieved in [86], which provided 3D morphodynamic simulations using a k-ω SST model, which was able to reproduce vortex shedding at the lee side of a surface mounted cylinder. Sumer [87], in his review on the mathematical modeling of scour, pointed out the free surface effects, the influence of small scale turbulence on sediment transport, and the potential effects of pore pressure on scour as the main avenues for future research. The study in [88] simulated scour around cylindrical and square piers, using a URANS (Unsteady Reynolds Averaged Navier Stokes) model coupled with an adapted version of the van Rijn model to estimate the bedload transport flux and the Exner equation. Better agreement was observed in the case where the shear drives the scour, as the HSV dynamics cannot be resolved with a URANS approach. Similar results were observed by [89], with the URANS and scour simulations past a surface mounted cylinder. This study showed that by considering the transport of suspended sediment, there is an improvement of the deposition patterns downstream of the pier. Recently, researchers in [90] proposed the use of a relaxation parameter to adjust the locally amplified bed shear stress due to the action of a horseshoe vortex to properly match the observed scour depth, solving the URANS–Exner equations system.

The work in [12,17,18] studied the dynamics of the HSV system at high-Reynolds numbers, using a hybrid URANS-LES turbulence model. These studies captured, for the first time, the intense velocity fluctuations and low-frequency bimodal oscillations, including the quasi-periodic vortex shedding and merging, and the formation of hairpin vortices, generated by a centrifugal instability that controls HSV dynamics. All these processes increase the instantaneous shear stress near the obstacle, allowing the development of models to study sediment dynamics from a Lagrangian perspective [23,91], as well as a model of bed evolution and bedform development, coupling the coherent-structure resolving model to the Exner equation [24], as shown in Figure 8a. These recent models have shown the potential of high-resolution 3D numerical simulations to predict different aspects of the scour process in subcritical conditions with good agreement. From these investigations, three key features have been identified: (1) Isotropic RANS turbulence models are inherently limited to predict the complex dynamics of an HSV system, as they yield a large turbulent viscosity, increasing the energy dissipation and suppressing the turbulence at the junction; (2) coherent structure resolving turbulence models can capture HSV dynamics, which can resolve the details of bedform formation and initial development of the scour hole in sand beds by turbulent structures; and (3) the main limiting assumption of these models stems from the prediction of instantaneous sediment fluxes from empirical formulas developed for uniform flows in steady and equilibrium conditions, adding ad-hoc formulations to represent the avalanches when the bed slope locally exceeds the angle of repose.

Using a Volume-Of-Fluid (VOF) approach, researchers [92] recently reported the first free surface resolving scour simulations using RANS–Exner type models. According to the author, accounting for the free surface effect improves the position of the HSV and the prediction of the mean velocity field, even at a Froude number as low as 0.2. The VOF technique emerges as a promising tool for the advanced numerical simulation of supercritical flows, including the scour at bridge foundations, following the coupled hydrodynamics and sediment transport approach.

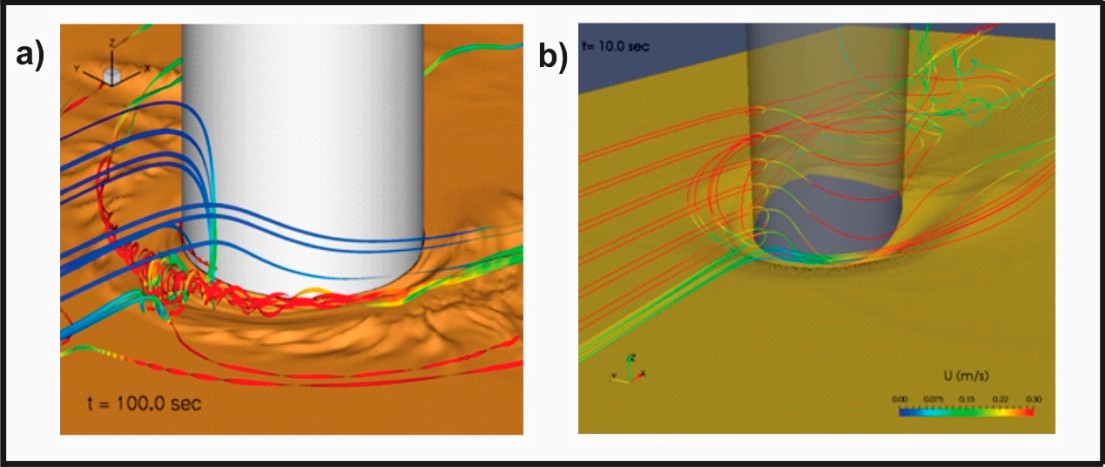

**Figure 8.** Instantaneous bed surface and streamlines around a cylindrical pier: (**a**) a subcritical flow with clear water conditions and the scour computed with the LES–Exner model, as in [24], and (**b**) a subcritical flow with live bed conditions computed with the turbulence averaged two-phase flow model, as in [93]. Streamlines are colored by the velocity magnitude.

### 4.4.2. Multi-Phase Flow Approach

Over the last decades, a new generation of sediment transport models has emerged: the Eulerian–Eulerian two-phase flow approach [94]. Unlike classical sediment transport models, the two-phase flow approach is based on the resolution of momentum balance for both the fluid and the sediment phases, the latter being seen as a continuum with a peculiar rheology. Very recently, using an open-source multi-dimensional two-phase flow model for sediment transport applications [95], the study in [93] presented the first two-phase flow RANS simulation for scour around a vertical cylinder. In Figure 8b, the scour mark induced by the HSV is clearly visible, while the downstream erosion induced by the lee–wake vortices was also observed. The two-phase flow approach is able to reproduce the scour dynamics induced by the HSV without empirical parametrization for the sediment fluxes and the avalanching process.

Further, the authors in [96] reported the first two-phase flow turbulence-resolving simulations. This opens new possibilities for simulating the complex interactions between sediment transport and HSV dynamics. Another important research possibility has been recently developed by [97] and concerns the development of a two-phase flow sediment transport model, including a free surface resolving capability. This model will allow one to reproduce the free surface features observed in the supercritical flow regime by [20,21] while solving sediment dynamics based on mechanical principles, as in [93].

Advances on scour modeling also consider computational techniques that efficiently calculate the dynamic coupling between the flow and the bed. RANS–Exner models coded in GPU (Graphics Processing Unit) have already been developed to improve the computation of these problems [98]. Future models that incorporate LES and Exner in the vicinity of bridge piers using these strategies show great promise in tackling large computational domains or fine resolutions in scour problems.

### 4.5. Moving from the Local Scale Phenomena Up to Long Term Dynamics

As detailed above, the scour process has been investigated mainly with laboratory and numerical approaches under several simplifying assumptions, neglecting the stochastic nature of floods. Indeed, the scour process should be considered a stochastic process controlled by the dynamics of floods, sediment transport, and riverbed evolution over time. All these processes act over a bridge foundation throughout the bridge's lifespan, and it is likely that the entire history of these events is responsible of the failure risk of a bridge rather than a specific flood event. An example of the potential time evolution

of flood characteristics (here: discharge and Froude number) and scour depth over time is given in Figure 9.

The limitation of the current knowledge of this phenomenon is highlighted in [99], which documented the magnitude of flood events causing bridge collapses in the US. Most of these bridges were designed for a flood event with a return period of about 100 years, but they collapsed under a variety of events with a return period ranging from 1 to 1000 years. This is clear evidence of the limitations of the current methodologies and understanding of bridge design. Lately, a number of authors have explored alternative strategies for bridge failure predictions in order to account for the non-stationarity of the flow and the stochastic nature of floods. These studies highlighted that scour evolution over time is strongly controlled by factors like the shape of flood hydrographs [83], and also the time-sequence of floods occurring at a given location [100]. Moreover, hydrological variability has been amplified even more significantly by the effects of global climate change [101].

Now, the interactions between infrastructure and river morphodynamics are characterized by many factors that span a wide range of temporal and spatial scales. Besides the hydrographs, additional factors at larger scales include variations in sediment availability, vegetation, land use in the watershed, channel management and the effects of anthropic interventions (e.g., gravel mining), or other infrastructures built in the channel, such as reservoirs or intakes. The sum of these phenomena should be considered when designing a bridge foundation, but actual modeling schemes, as well as available information, are not able to fully describe such a complex dynamic. Therefore, the challenge in the coming years is to identify unifying principles able to interpret scour phenomena, both under subcritical and supercritical conditions, with the right compromise between model complexity and data availability [102–104]. Moreover, it is highly desired that forthcoming modeling approaches explicitly account for the stochastic nature of the flood process.

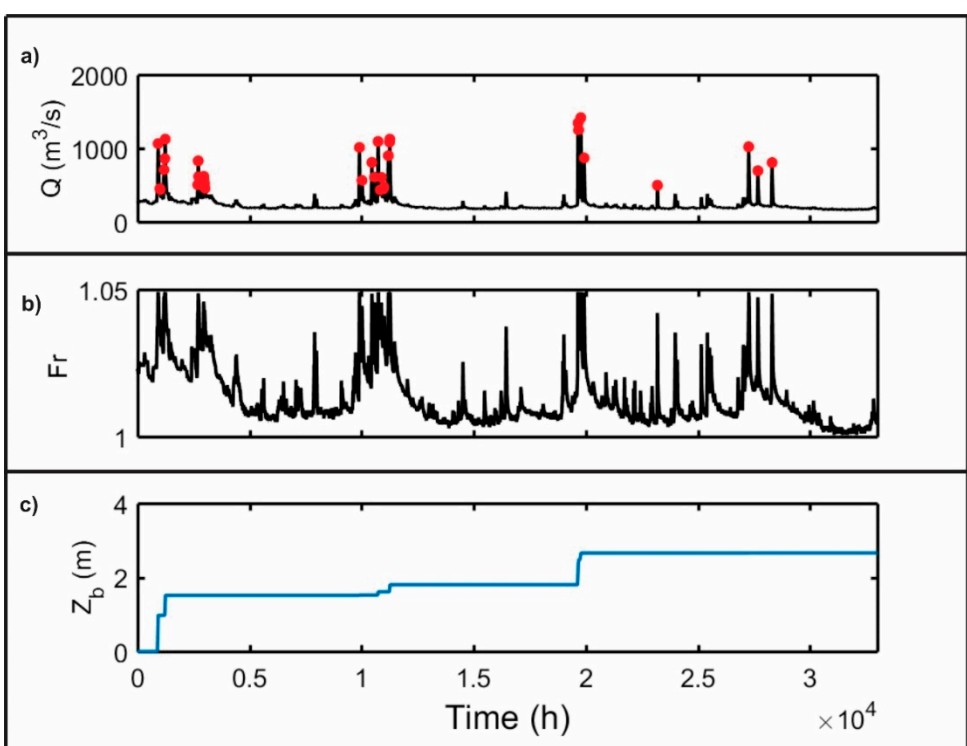

**Figure 9.** The synthetic time-series over a temporal window of approximately 4 years: (**a**) supercritical flow discharge, (**b**) the corresponding Froude number Fr, and (**c**) the evolution of the scour depth at the foot of a bridge pile, assuming it as an additive process. The graph displays the random nature of the flood events within a river system and its consequences in terms of the random evolution of scour depth over time, computed using the BRISENT model, as in [105].

## 5. Discussion and Conclusions

Bridges with foundations in river beds with supercritical flow regimes are frequent and can be found worldwide. Handling the scouring process that takes place in this type of flow (e.g., with bed sills [106]) is crucial for the structural integrity of bridges, transportation systems, and the safety of users. Scour around piers overlaps with several processes occurring at different spatial and temporal scales, such as bed degradation and aggradation, dynamic braiding or meandering, and the formation/destruction of antidunes, large woody debris [69], barbs [107–109], or spur dikes for preserving the desired water depth [110]. Moreover, extreme flood events produce rapid changes of the local riverbed morphology, affecting the scouring process at bridge foundations. The temporal dynamics of scour, then, are intimately linked with sediment transport. This paper focused on the knowledge, gaps, and open questions related to local scour at bridge foundations in supercritical flows. From our review, the controlling mechanisms of scour in supercritical flows differ from those controlling scour in subcritical flows, due to the appearance of a detached hydraulic jump or vertical wall-jet patterns [20]. Interactions with the horseshoe and wake vortex systems (and also with the sediment motion) remain nearly unexplored (except for the work in [21]). The unexpected scour depth in the available experiments [6,27] suggests that the maximum scour depth in supercritical flows might be of comparable magnitude to that occurring in subcritical flows, even though the flow intensity is much higher. Further research is needed to understand the controlling parameters of the scour in supercritical flows.

### 5.1. Measuring Techniques for Flow in the Laboratory

Laboratory scour experiments with supercritical flows should avoid intrusive measuring techniques due to their important effect on the flow field (especially creating surface waves). Efforts should be made to transfer some widely used optical techniques from fluid mechanics to the hydraulics of supercritical flows, such as high speed image based velocimetry, including two dimensional and stereo PIV/PTV, or even the use of tomographic velocimetry, taking advantage of techniques like index matching to access the flow field in the scour hole.

### 5.2. Measuring Techniques for Scour in the Laboratory

Scour hole geometry might be best measured with precision ultrasonic or optic distance sensors. However, important constraints appear for intrusive instruments, such as sonars, especially at the laboratory scale because the size of the instrument can alter the flow field and create surface waves. Thus, the use of video cameras placed inside transparent bridge elements is recommended to record the maximum scour depth at the front of the bridge foundation over time.

### 5.3. Measuring Techniques for Flow in the Field

Field measurements of the supercritical flow at bridges might provide a good idea of the flow velocity upstream of the impacted foundations. Accessibility is, by far, identified as the main constraint in the use of intrusive equipment, so radar velocimeters [70] and video analysis, such as LSPIV [71], are recommended. As these methods provide only the surface velocity, further research is needed to correlate this data with the corresponding flow field and features.

### 5.4. Measuring Techniques for Scour in the Field

A major issue for river engineers dealing with bridge foundation design is to evaluate all the processes that could affect erosion for a better assessment of the scouring intensity and, consequently, a better optimization of the bridge foundation cost. Three main processes affecting the bed level generally include (i) the overall long-term bed evolution linked to the river equilibrium (indeed, many rivers in the world suffers from erosion due to gravel mining or damming [111]); (ii) the natural river breath during a flood, leading to global erosion during the rising part of the flood and global deposition

during the ebb part of the flood; and (iii) local erosion at the bridge foundation. Eventually, long term evolution should be tackled by being able to estimate the maximum erosion during an event but also to estimate the refilling of the scour after this event. Monitoring of very dynamic rivers remains quite difficult but obviously needs some investment to be able to verify the upscaling limits from laboratory experiments. The measurement of the local scour depth at real bridges in supercritical flows or during floods has succeeded until now by only using the numbered brick method [5,75]. Further research is needed to develop methods using the structural behavior of the bridge for scour measurement. These kinds of indirect methods would allow for scour monitoring in complex hydro-sedimentary conditions.

### 5.5. Physical Modeling of Scour in Supercritical Flows

Given the constraints of knowledge transfer from sub to supercritical scour, and the important difficulties for laboratory and field measurements of flow and scour at bridge foundations, the use of physical scale models appears to be a reasonable alternative for the analysis and design of particular cases. Important requirements include a wide set-up to prevent lateral confinement effects, high water and sediment discharges, and innovative measuring techniques in line with the involved rapid velocities. Moreover, important scale effects are expected in the case of model sediments with mean diameters smaller than 0.2 mm.

### 5.6. Numerical Modeling of Scour in Supercritical Flows

Classical coupled Navier-Stokes-Exner models have had some success in predicting the scour around bridge piles under subcritical flow conditions, even though some open questions remain to be answered. However, to the best of the authors' knowledge, no simulation of scour under supercritical flow conditions has ever been reported in the literature. Such simulations would be very challenging, particularly because of the very energetic flow conditions encountered in supercritical flows, the strong free surface dynamics, the multiple feedback mechanisms between the free surface dynamics, the flow hydrodynamics (e.g., HSV), and the sediment dynamics or the air entrainment. Some important breakthroughs can be expected in the near future by using multi-phase flow approaches in conjunction with high-resolution experimental data. From a numerical modeling standpoint, capturing the dynamics of a free surface is a critical prerequisite for numerical simulations of supercritical flows. To resolve the coupled flow field and interactions between the large-scale coherent structures of the HSV system and the free-surface, LES, or hybrid URANS–LES turbulence models should be tested to understand the flow physics and unsteady scour mechanisms. Eulerian–Lagrangian approaches seems unrealistic due to their very high computational cost. However, turbulence-resolving Eulerian–Eulerian simulations coupled with a free surface resolving capability will probably be possible in the near future.

**Author Contributions:** All eight authors (O.L., E.M., S.R., B.C., C.E., J.C., W.B. and S.M.) contributed to all aspects of present work in the framework of a one-week workshop at the University of Concepcion, Chile.

**Funding:** Financial support by the Chilean research council CONICYT through grants Fondecyt 1150997 and REDES 170021 is greatly acknowledged. C.E. acknowledges the support of Conicyt/Fondap grant 15110017.

**Acknowledgments:** The authors would like to thank Nicolas Rivière (University of Lyon) for his scientific advice.

**Conflicts of Interest:** The authors declare no conflict of interest.

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
