# Peer review of "Scour at Bridge Foundations in Supercritical Flows: An Analysis of Knowledge Gaps"

_water, doi:10.3390/w11081656_

Round 1

Reviewer 1 Report

This is an extensive and well-written review article on the problem of bridge scour. The paper delivers as promised in the abstract and introduction, and the conclusion and recommendations are well founded. I recommend publication as is, I only found one tiny spelling error: "technics" on line 274, page 9, should probably be "techniques".

Author Response

This is an extensive and well-written review article on the problem of bridge scour. The paper delivers as promised in the abstract and introduction, and the conclusion and recommendations are well founded. I recommend publication as is, I only found one tiny spelling error: "technics" on line 274, page 9, should probably be "techniques".

This modification was included in the revised manuscript

Reviewer 2 Report

Some more original experimental results could be appreciated

Author Response

Some more original experimental results could be appreciated

The authors fully concur with the reviewer: more experimental works dedicated to scour in supercritical conditions is required to better understand the governing processes and propose design guidelines. Several co-authors of this paper are currently preparing proposals to further investigate this topic.

Reviewer 3 Report

This manuscript presents a review paper on scouring at bridge foundation under supercritical flows. The introduction is well-written and sets an appropriate context. The rest of the sections are well-described and organized. However, I feel there are still details to be incorporated in the manuscript. I therefore provide a short list of comments below.

Comments:

1.       In subsection 4.1  RGB-D sensors used to measure continuously the free surface should be mentioned. These low-cost 3D sensors have appeared as a consequence of the videogame industry development. The original devices, Carmine (released by Primesense) and Kinect (released by Microsoft in 2010), were designed to capture human body movements for interaction with video games. By analyzing such information depth information can be extracted. These devices have been successfully used for measuring the surface of dry granular flows in in [1] and the surface of gravity waves in clear water [2].

2.       I would add a subsection discussing the similarities/dissimilarities of flume experiments compared to field studies. Further, the challenges in providing scalability (in terms of hydraulics and sediment erosion) between flume experiments and field studies should be exposed.

3.       In subsection 4.4. I would add as a knowledge gap the possibility of coupling RANS-EXNER models coded in GPU [3] (i.e. models which computationally are efficient) with LES-EXNER models in the vicinity of bridge piers. This strategy will increase the efficiency of large computational domains which may include bridges.

 [1] Caviedes-Voullième, D. , Juez, C. , Murillo, J. , García-Navarro, P. , 2014. 2D dry granular free-surface flow over complex topography with obstacles. part I: experimental study using a consumer-grade RGB-D sensor. Computers & Geosciences 73 (0), 177–197 .

[2] Nichols, A. , Rubinato, M. , 2016. Remote sensing of environmental processes via low-cost 3d free-surface mapping. In: Proceedings of the 4th IAHR Europe Congress .

[3] C Juez, A Lacasta, J Murillo, P García-Navarro. An efficient GPU implementation for a faster simulation of unsteady bed-load transport. 2016. JOURNAL OF HYDRAULIC RESEARCH 54 (3), 275-288.

Author Response

1.       In subsection 4.1  RGB-D sensors used to measure continuously the free surface should be mentioned. These low-cost 3D sensors have appeared as a consequence of the videogame industry development. The original devices, Carmine (released by Primesense) and Kinect (released by Microsoft in 2010), were designed to capture human body movements for interaction with video games. By analyzing such information depth information can be extracted. These devices have been successfully used for measuring the surface of dry granular flows in in [1] and the surface of gravity waves in clear water [2].

RGB-D sensors are mentioned in the section dedicated to the “Image-based reconstruction of free surface » methods. The following text was added: “On the other hand, the access of the instantaneous 2D free-surface deformation around the pier/abutment requires projection techniques such as the RGB-D sensors method [60], the projected grid method [27] with a limited spatial resolution, or the Fourier transform profilometry method [61] “

2.       I would add a subsection discussing the similarities/dissimilarities of flume experiments compared to field studies.

Unfortunately, very limited information exists today from field studies of scour due to supercritical flows. This is the reason why section “3. Scour around bridge foundations” contains no subsection dedicated to field studies (it contains only two subsections related to laboratory flows). To date the main information available regarding field studies are damaged bridge piles observed after the supercritical flow event.

2b       Further, the challenges in providing scalability (in terms of hydraulics and sediment erosion) between flume experiments and field studies should be exposed.

Section 4.3 is dedicated to discuss the challenge of scalability of scour processes in laboratory studies of subcritical scour processes. We discuss the challenges to reproduce a flume at scale with real scouring processes taking place in the field: sediment scaling, problem of sediment cohesion, viscous effects as the Reynolds number is not maintained equal between the field case and the laboratory experiments

Moreover, in the supercritical conditions, additional technical challenges arise in terms of high sediment and water discharges to be regularly supplied in live-bed experiments and the technical challenges in terms of metrology for measuring the inflow velocity, maximum scour depth and the shape of the scour hole. However, the authors feel that we lack much information to identify the major limitations of such scour in supercritical flow experiments. For this reason, we included the following sentence: “Still, verifications of these scaling laws for the supercritical case are needed before such scaling approach can be employed with confidence in physical modeling”

3.       In subsection 4.4. I would add as a knowledge gap the possibility of coupling RANS-EXNER models coded in GPU [3] (i.e. models which computationally are efficient) with LES-EXNER models in the vicinity of bridge piers. This strategy will increase the efficiency of large computational domains which may include bridges.

The following text was added to the revised manuscript: “Advances on scour modeling can also consider computational techniques that calculate efficiently the dynamic coupling between the flow and the bed. RANS-Exner models coded in GPU, have already been developed to improve the computation of these problems [98]. Future models that can incorporate LES and Exner in the vicinity of bridge piers using these strategies might show great promise to tackle large computational domains or fine resolutions in scour problems.”

Reviewer 4 Report

The authors present a review paper on scour at bridge foundations in supercritical flows, summarizing the results of existing literature on flow and scour mechanisms and describing available measuring techniques.

The title of the paper is very promising, the quality of the writing is generally good, but at the end of the reading I felt a bit disappointed, as I mainly found only a superficial review of previous works on the topic, without a very in depth discussion and analysis of the problems at stake. In addition, recommendations provided in the paper on suitable measurement techniques are also quite obvious for a readership involved in experimental modelling of flow and scour processes in channels.

Author Response

The title of the paper is very promising, the quality of the writing is generally good, but at the end of the reading I felt a bit disappointed, as I mainly found only a superficial review of previous works on the topic, without a very in depth discussion and analysis of the problems at stake. In addition, recommendations provided in the paper on suitable measurement techniques are also quite obvious for a readership involved in experimental modelling of flow and scour processes in channels.

The aim of the present paper is indeed to expose a literature review on hydrodynamics and scour in supercritical flows around bridge piers and to draw a research perspective on this important topic left apart by most of the community. Regarding the recommendations for lab and physical model measurements, we intended to assess the challenges of access to flow characteristics is such a rapid and opaque flow with strong surface deformation. We reviewed the available in the community and discussed their applicability in the specific case of supercritical regime.

Round 2

Reviewer 2 Report

The authors declare that the paper is preliminary to other ecperimental eorks.

Perhaps the same affirmation could be introduced in the paper text.

Reviewer 4 Report

The comments are the same as in Report 1. I leave the final decision to the editor.